

# A novel bicyclic 2,4-diaminopyrimidine inhibitor of *Streptococcus suis* dihydrofolate reductase

Warangkhana Songsungthong[1], Sunisa Prasopporn[1,2], Louise Bohan[1,3], Potjanee Srimanote[4], Ubolsree Leartsakulpanich[1] and Suganya Yongkiettrakul[1]

[1] Biosensing and Bioprospecting Research Group, National Center for Genetic Engineering and Biotechnology (BIOTEC), National Science and Technology Development Agency (NSTDA), Pathum Thani, Thailand
[2] Current Address: Department of Pharmacology, Faculty of Medicine, Siriraj Hospital, Mahidol University, Bangkok, Thailand
[3] Current Address: School of Pharmacy and Life Sciences, The Robert Gordon University, Aberdeen, United Kingdom
[4] Faculty of Allied Health Sciences, Thammasat University, Klong Luang, Pathum Thani, Thailand

Corresponding author
Warangkhana Songsungthong,
warangkhana.son@biotec.or.th

## ABSTRACT

*Streptococcus suis* is a Gram-positive bacterial pathogen of pigs and an emerging zoonotic pathogen. It has become increasingly resistant to multiple classes of antibiotics. New drug candidates and knowledge of their targets are needed to combat antibiotic-resistant *S. suis*. In this study, the open-source Pathogen Box compound library was screened. Thirty hits that effectively inhibited *S. suis* growth at 10 μM were identified. Among the most potent hits, MMV675968 (a diaminoquinazoline analog) was shown to target *S. suis* dihydrofolate reductase (*Ss*DHFR) via (1) growth inhibition of an *E. coli* surrogate whose growth is dependent on exogenously expressed *Ss*DHFR and (2) inhibition of in vitro *Ss*DHFR activity. Thymidine supplement is able to reverse growth inhibition by MMV675968 in both *E. coli* surrogate and *S. suis*, indicating that a thymidine-related pathway is a major target of MMV675968. Comparison of MMV675968 with seven DHFR inhibitors representing different core structures revealed that bicyclic 2,4-diaminopyrimidines with long and flexible side chains are highly effective in inhibiting *Ss*DHFR and *S. suis* growth. MMV675968 and related compounds thus may serve as starting points for developing antibiotics against drug resistant *S. suis*.

## INTRODUCTION

It is estimated that antibiotic resistance will lead to 10 million deaths per year and economic loss of US$100 trillion by the year 2050 (*O'Neill, 2014*). Alarmingly, bacteria resistant to last resort antibiotics and to all commercially available antibiotics have already emerged and spread, signaling the start of a global public health crisis (*McGann et al., 2016*; *Ruzauskas & Vaskeviciute, 2016*; *McCarthy, 2017*). New antibiotics, preferably with new chemical scaffolds to bypass existing resistance mechanisms, are urgently needed. Pathogen Box, an open-source library of 400 compounds with chemical structures distinct from currently

available antibiotics (https://www.mmv.org/mmv-open/pathogen-box/about-pathogen-box), demonstrated activity against selected neglected disease pathogens and low toxicity against human cells is a promising source of compounds for antibiotic discovery.

*Streptococcus suis* is a Gram-positive bacterium that can cause severe symptoms such as meningitis, septicemia, and arthritis in pigs and thus pose a high economic burden for the global pig industry (*Wertheim et al., 2009a*). Zoonotic transmission to humans occurs from eating uncooked contaminated pork or from occupation-related infection such as when farmers or abattoir workers come into contact with infected pigs or pig carcasses (*Wertheim et al., 2009a*; *Ho et al., 2011*; *Goyette-Desjardins et al., 2014*; *Huong et al., 2014*). *S. suis* isolates found in pigs and in human cases in Vietnam and China are of the same serotype and multi locus sequence typing (MLST), suggesting direct transmission (*Hoa et al., 2011*). In human infection, *S. suis* causes meningitis, sepsis, arthritis, vestibular dysfunction, and permanent hearing loss with fatality rate of approximately 13% (*Huong et al., 2014*). *S. suis* infection is a major cause of bacterial meningitis in Southeast Asia, and was responsible for a Streptococcal Toxic Shock Syndrome outbreak in China (*Suankratay et al., 2004*; *Hui et al., 2005*; *Tang et al., 2006*; *Thi Hoang Mai et al., 2008*; *Wertheim et al., 2009b*). The number of human *S. suis* cases has increased recently, especially in areas with high density pig farming, making *S. suis* infection an important emerging disease (*Wertheim et al., 2009a*; *Gajdács et al., 2020*).

Owing to widespread use of antibiotics for prophylaxis and as growth promoter in the pig farming industry, antibiotic resistance in *S. suis* is rising (*Wertheim et al., 2009a*; *Varela et al., 2013*; *Hernandez-Garcia et al., 2017*; *Yongkiettrakul et al., 2019*). *S. suis* isolates are recently reported to be resistant to multiple classes of antibiotics such as $\beta$-lactams (e.g., ampicillin and penicillin), macrolides (e.g., erythromycin and clarithromycin), lincosamide (e.g., clindamycin), tetracycline, fluoroquinolone (e.g., moxifloxacin), aminocyclitol (e.g., spectinomycin), aminoglycoside (e.g., gentamycin), and trimethoprim-sulfonamide (*Yongkiettrakul et al., 2019*; *Tan et al., 2020*; *Riley et al., 2020*; *Gajdács et al., 2020*). Multidrug resistance in *S. suis* thus prompts the need for new drugs. In this study, we screened the Pathogen Box library against two strains of *S. suis*, namely P1/7 and HE06 (*Jacobs, Van den Berg & Loeffen, 1996*; *Maneerat et al., 2013*) and identified 30 compounds with high growth inhibitory activity at 10 µM. The potential target of MMV675968, a diaminoquinazoline derivative, was identified as dihydrofolate reductase, which is a target of multiple classes of compounds including antibacterial, antimalarial, and anticancer compounds (*Gao et al., 2019*).

## MATERIALS AND METHODS

The experiments using *S. suis* were approved by BIOTEC Institutional Review Board on biosafety and biosecurity with approval number BT-IBC-59-028.

### Bacteria strains and growth conditions

*S. suis* strains P1/7 and HE06 are serotype 2 isolates obtained from a pig with meningitis and an infected human, respectively (*Jacobs, Van den Berg & Loeffen, 1996*; *Holden et al., 2009*; *Maneerat et al., 2013*). Serotype 2 strains were chosen because of their association

with pathogenicity (*Lun et al., 2007*; *Wertheim et al., 2009a*; *Maneerat et al., 2013*). *S. suis* was grown on brain heart infusion (BHI) plates (Beckton Dickinson, Franklin Lakes, NJ, USA) at 37 °C with 5% $CO_2$. *Escherichia coli* PA414 lacking *folA* and *thyA* genes, encoding dihydrofolate reductase (DHFR) and thymidylate synthase (TS), respectively (*Ahrweiler & Frieden, 1988*) was used as a surrogate for expressing *S. suis* DHFR. The deletion of *thyA* is required for survival of DHFR deficient mutant (*Ahrweiler & Frieden, 1988*). *E. coli* PA414 transformed with various expression plasmids was grown in Luria Bertani (LB) broth supplemented as necessary with 50 µg/mL kanamycin, 100 µg/mL ampicillin, 10 µg/mL chloramphenicol, 0.2% (w/v) arabinose, and/or 50 µg/mL thymidine.

## Bacterial growth inhibition assay

The Clinical Laboratory Standard Institute (CLSI) broth microdilution method was used to test antibacterial activity of Pathogen Box compounds and selected DHFR inhibitors (*The Clinical and Laboratory Standards Institute, 2018*). Briefly, $5 \times 10^4$ colony forming units (CFUs) of *S. suis* or *E. coli* PA414 were incubated with compounds at 10 µM (or other concentrations as indicated) in cation-adjusted Mueller Hinton broth (Beckton Dickinson, Franklin Lakes, NJ, USA) in 96-well plates at 37 °C with or without 5% $CO_2$, respectively. Growth was monitored by measuring optical density at 600 nm ($OD_{600}$) using a SpectraMax M5 spectrophotometer (Molecular Devices, San Jose, CA, USA). Percent growth of each compound-treated bacteria strain was calculated using the following formula:

(OD of compound treated bacteria $\times$ 100)/(OD of DMSO treated bacteria).

## Construction of expression plasmids

The *dhfr* gene encoding *S. suis* dihydrofolate reductase (*Ss*DHFR) was amplified from clarified lysate of *S. suis* P1/7 and *S. suis* HE06 using high-fidelity Phusion DNA polymerase (New England Biolabs, Ipswich, MA, USA) and primers 5′ctggtgccgcgcggcagccatATGACTAAAAAGATTGTTGC3′ and 5′tcgggctttgttagcagccggatc CTAACCATCTCTTCTTTCATAG3′ (lower case denotes sequence homologous to pET15b while upper case denotes sequence homologous to the *dhfr* gene). The PCR products were cloned into NdeI and BamHI digested pET15b plasmid using a Gibson Assembly kit (New England Biolabs, Ipswich, MA, USA), according to manufacturer's protocol. The resulting plasmids were subjected to Sanger sequencing (1st BASE, Singapore). Nucleotide sequences of the *dhfr* gene from *S. suis* P1/7 and *S. suis* HE06 were deposited in NCBI Genbank with accession number MH388486 and MH388487, respectively.

## Complementation assay of *E. coli* surrogate

*E. coli* PA414 lacking functional DHFR and TS enzymes (*Ahrweiler & Frieden, 1988*) was used as a surrogate host to study the function of exogenously expressed *Ss*DHFR. *E. coli* PA414 was co-transformed with (1) pBAD33 or pBAD33-*Ec*TS and (2) pET15b or pET15b-*Ss*DHFR. To test whether *Ss*DHFR can complement for the loss of *E. coli* DHFR in the *E. coli* surrogate, overnight cultures of *E. coli* PA414 carrying various combination of plasmids were pelleted, washed, cell density adjusted to $OD_{600}$ of 4, serially-diluted, and

spotted onto LB plates supplemented with appropriate antibiotics and either 50 µg/mL thymidine or 0.2% (w/v) arabinose. Plates were incubated at 37 °C overnight. Growth on agar plate was observed.

### Overexpression of *Ss*DHFR in *E. coli* BL21(DE3)

Cultures of *E. coli* BL21(DE3) carrying pET15b or pET15b-*Ss*DHFR were grown with shaking at 37 °C in LB supplemented with 100 µg/mL ampicillin. When $OD_{600}$ reached 1, IPTG was added to 40 µM, temperature shifted to 16 °C, and incubated with shaking overnight. Bacterial pellets were collected by centrifugation, resuspended in lysis buffer (20 mM potassium phosphate buffer, 0.1 mM EDTA, 10 mM DTT, 50 mM KCl, 20% glycerol), and sonicated. Protein concentration of clarified bacteria lysate was determined by Bradford assay (Biorad, Hercules, CA, USA) according to the manufacturer's protocol.

### In vitro DHFR activity assay

DHFR uses NADPH as a cofactor for the conversion of dihydrofolate (DHF) to tetrahydrofolate (THF). Since the amount of NADPH consumed is directly proportional to the amount of THF product, monitoring NADPH consumed using absorbance at 340 nm ($A_{340}$; $\varepsilon_{340}$ of 12,300 $M^{-1}$ $cm^{-1}$) is indicative of DHFR enzymatic activity. The DHFR activity assay was performed as previously described (*Tirakarn et al., 2012*; *Songsungthong et al., 2019*). Briefly, using kinetics mode of a UV-Vis spectrophotometer (Agilent Technologies, Santa Clara, CA, USA), $A_{340}$ value of the reaction mixture (50 mM TES, pH 7.0, 75 mM $\beta$-mercaptoethanol, 1 mM EDTA, 1 mg/mL BSA, 0.1 mM NADPH, 0.1 mM dihydrofolate) was set to blank ($A_{340} = 0$). Inhibitors were added to the reaction mixture as necessary. 1 µg of total protein from lysate of *E. coli* BL21(DE3) containing pET15b or overexpressing *Ss*DHFR was added to reaction buffer to initiate the enzymatic reaction. The amount of *Ss*DHFR containing lysate added was predetermined to give linear reaction kinetics. NADPH reduction was monitored by measuring $A_{340}$ reduction for 100 s. $A_{340}$ reduction per minute values were recorded.

Percent *Ss*DHFR activity was calculated using the following formula:

($A_{340}$ reduction per minute of compound-treated lysate $\times$ 100)/($A_{340}$ reduction per minute of DMSO-treated lysate of BL21 expressing *Ss*DHFR).

### Statistical analysis

Data are shown as mean $\pm$ standard error of the mean (SEM). At least three independent experiments were performed. Statistical analysis using one-way ANOVA with Tukey's post test was performed. Statistical significance is noted on the graph by asterisks. * denotes $P < 0.05$. ** denotes $P < 0.01$. ** denotes $P < 0.001$. *** denotes $P < 0.0001$. ns denotes not statistically significant.

## RESULTS

### Identifying compounds with *S. suis* growth inhibitory activity from Pathogen Box

We screened the Pathogen Box library for compounds active against *S. suis* P1/7 and *S. suis* HE06 isolated from an infected pig and human, respectively (*Jacobs, Van den Berg*

*& Loeffen, 1996*; *Holden et al., 2009*; *Maneerat et al., 2013*). Thirty compounds with high inhibitory activity (with average percent inhibition of 90–100%) at 10 µM were identified (Table S1). Seven of the hits including rifampicin, levofloxacin, and linezolid are reference compounds known to be broadly effective against Gram-positive bacteria (highlighted in blue in Table S1). Twenty three other hits are compounds with structures distinct from commercially available antibiotics (highlighted in green in Table S1). The mechanisms of action of most non-reference hits are unknown. MMV675968, one of the hits, has been shown to inhibit the growth of protozoan parasites and Gram-negative bacteria (*Lau et al., 2001*; *Nelson & Rosowsky, 2001*; *Popov et al., 2006*; *Songsungthong et al., 2019*) via inhibition of dihydrofolate reductase (DHFR), an enzyme in the thymidylate cycle.

## MMV675968 is an inhibitor of *S. suis* DHFR

MMV675968 inhibits DHFR of protozoan parasites and of Gram-negative bacteria (*Lau et al., 2001*; *Nelson & Rosowsky, 2001*; *Popov et al., 2006*; *Songsungthong et al., 2019*). We therefore hypothesized that MMV675968 inhibited *S. suis* growth through inhibition of *S. suis* DHFR (*Ss*DHFR). An *E. coli* surrogate assay and an in vitro DHFR activity assay were used to test this hypothesis. For the *E. coli* surrogate assay, *E. coli* PA414 deficient in *E. coli* DHFR (*Ec*DHFR) and *E. coli* thymidylate synthase (*Ec*TS) of the thymidylate cycle making the strain a thymidine auxotroph (*Ahrweiler & Frieden, 1988*), was used as a host for exogenously expressing *Ss*DHFR. *E. coli* PA414 transformant controls carrying two empty vectors (pET15b and pBAD33), or expressing *Ec*TS alone did not grow in the absence of thymidine supplement as expected (Fig. 1A). *E. coli* PA414 expressing *Ec*TS together with *Ss*DHFR was able to grow without thymidine supplement (Fig. 1A), indicating that *Ss*DHFR was functional and complemented for the loss of *Ec*DHFR. Growth of *E. coli* PA414 in the absence of thymidine was therefore dependent on the function of exogenously expressed *Ss*DHFR.

Trimethoprim (TMP), a bacterial DHFR inhibitor (*Baker et al., 1981*), inhibited the growth of *E. coli* PA414 expressing *Ss*DHFR to 19% of untreated (Fig. 1B), confirming that trimethoprim inhibits *Ss*DHFR. MMV675968 was more effective than TMP as seen by its ability to inhibit the growth of *E. coli* PA414 expressing *Ss*DHFR to 7% (Fig. 1B). *E. coli* PA414 growth inhibition by MMV675968 was reversed in the presence of thymidine supplement (Fig. 1C), suggesting that the thymidine-related pathway is the principal target of MMV675968 in *E. coli* surrogate consistent with DHFR being a target.

Next, we tested whether MMV675968 could inhibit *Ss*DHFR activity directly by an in vitro DHFR activity assay. DHFR activity present in lysate of *E. coli* BL21(DE3) carrying an empty pET15b plasmid accounted for only 2% of DHFR activity of lysate of *E. coli* BL21(DE3) overexpressing *Ss*DHFR (Fig. 1D), indicating that most of the DHFR activity observed was from overexpressed *Ss*DHFR rather than endogenous *Ec*DHFR. In the presence of 1 µM MMV675968, DHFR activity decreased to 2% of that of untreated lysate (Fig. 1D), indicating that MMV675968 inhibits *Ss*DHFR directly. Results from both *E. coli* surrogate assay and DHFR activity assay thus indicate that MMV675968 inhibits *Ss*DHFR.

We also tested whether thymidine-related pathway is the main target of MMV675968 in *S. suis* by testing whether growth inhibition was rescued by thymidine supplement. Growth

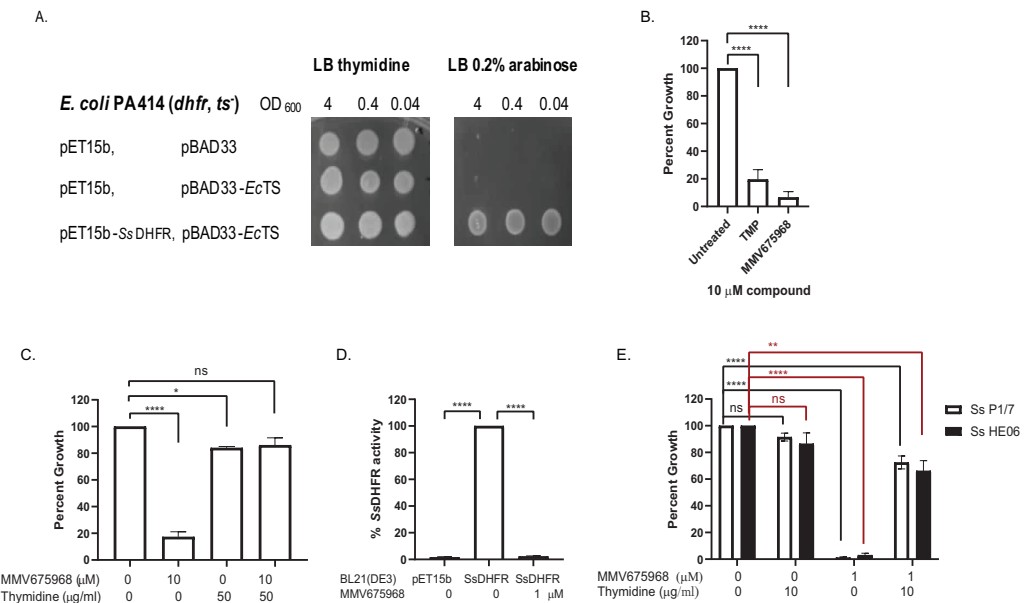

**Figure 1** **MMV675968 inhibits *Streptococcus suis.* dihydrofolate reductase (*Ss* DHFR).** (A) *Ss* DHFR functions in *E. coli* PA414. Overnight cultures of *E. coli* PA414 carrying empty plasmids or expressing *Ec*TS and *Ss*DHFR were serially diluted and spotted onto LB plates supplemented with either 50 μg/mL thymidine or 0.2% (w/v) arabinose. Overnight growth was recorded. A representative picture from three independent experiment is shown. (B) MMV675968 inhibits *Ss*DHFR in *E. coli* surrogate. *E. coli* PA414 expressing *Ec*TS and *Ss*DHFR was incubated with DMSO, trimethoprim (TMP), or MMV675968 (10 μM). Growth of compound-treated *E. coli* surrogate was calculated as percent of untreated (100%). (C) Growth inhibition of *E. coli* surrogate expressing *Ss*DHFR by MMV675968 can be rescued by thymidine supplement. *E. coli* PA414 expressing *Ec*TS and *Ss*DHFR was incubated with or without 10 μM MMV675968 and with or without 50 μg/mL thymidine. Growth was monitored and compared with that of untreated control (100%). (D) MMV675968 inhibits *Ss*DHFR activity. *In vitro Ss*DHFR activity from clarified lysate of *E. coli* BL21(DE3) overexpressing *Ss*DHFR was compared with that of *E. coli* BL21(DE3) harboring empty plasmid or in the presence of 1 μM MMV675968. (E) *S. suis* growth inhibition by MMV675968 can be rescued by thymidine supplement. *S. suis* P1/7 and *S. suis* HE06 was incubated with or without 1 μM MMV675968 and with or without 10 μg/mL thymidine. Growth was monitored and compared with that of untreated control (100%). The graphs (B–E) show mean ± standard error of the mean (SEM) from three independent experiments. One-way ANOVA with Tukey's post test was performed to determine statistical significance compared with untreated control. *, $P < 0.05$; **, $P < 0.01$; ****, $P < 0.0001$; ns, not statistically significant.

of both *S. suis* P1/7 and *S. suis* HE06 were inhibited in the presence of 1 μM MMV675968 (Fig. 1E). Growth inhibition of both *S. suis* strains was reversed with thymidine supplement (Fig. 1E), indicating that the main target of MMV675968 within *S. suis* is a thymidine-related pathway.

## Bicyclic 2,4-diaminopyrimidines with long and flexible side chain associate with higher inhibitory activity against *Ss*DHFR and *S. suis*

To investigate which chemical scaffolds are better at inhibiting *Ss*DHFR and *S. suis* growth, eight DHFR inhibitors of various chemical structures (Figs. 2A–2C), which include commercially available compounds (pemetrexed, trimethoprim, pyrimethamine, cycloguanil, and methotrexate), compounds from Medicines for Malaria Venture (MMV)'s

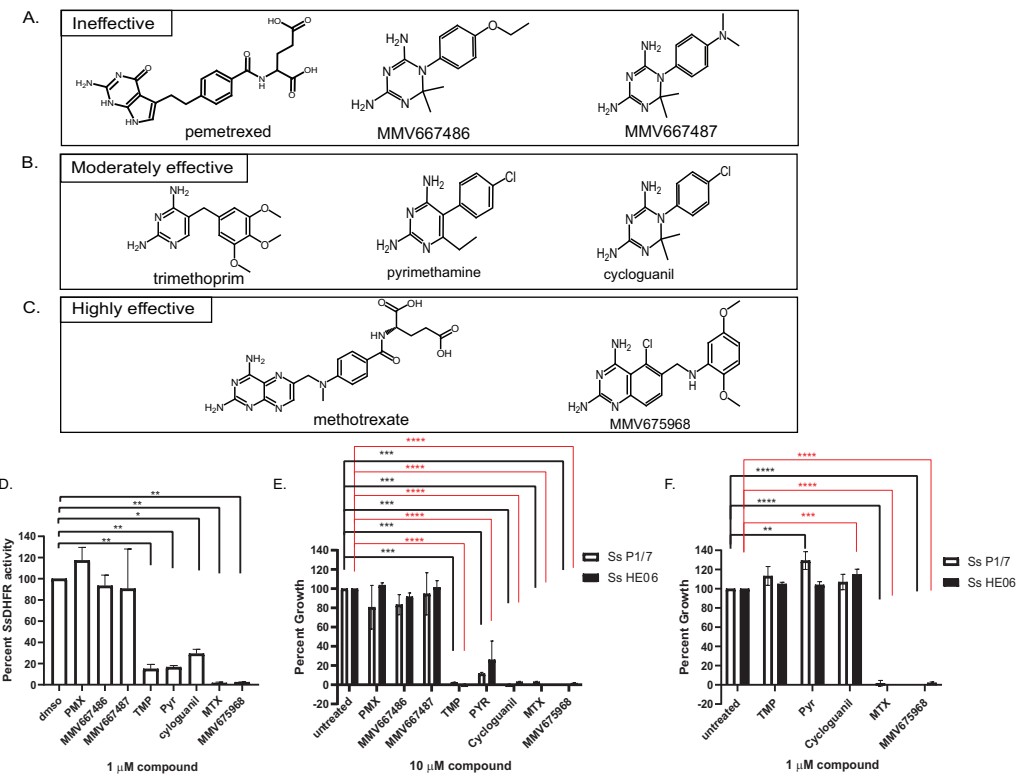

**Figure 2** **Efficacy of various dihydrofolate reductase (DHFR) inhibitors of *Streptococcus suis* DHFR (*Ss*DHFR) and *S. suis* growth.** Structures of eight DHFR inhibitors tested in this study and grouped according to their inhibitory activity: (A) ineffective inhibitors, (B) moderately effective inhibitors, and (C) highly effective inhibitors. (D) *Ss*DHFR activity in the presence of 1 µM DHFR inhibitors. (E) Percent growth of *S. suis* P1/7 (white bar) and *S. suis* HE06 (black bar) after 18 h incubation with 10 µM of DHFR inhibitors. (F) Percent growth of *S. suis* P1/7 (white bar) and *S. suis* HE06 (black bar) after 18 h incubation with 1 µM inhibitors. PMX, pemetrexed; TMP, trimethoprim; PYR, pyrimethamine; MTX, methotrexate. The graphs show mean ± standard error of the mean (SEM) from at least three independent experiments. One-way ANOVA with Tukey's post test was performed to determine statistical significance compared with untreated control. *, $P < 0.05$; **, $P < 0.01$; ***, $P < 0.001$; ****, $P < 0.0001$; ns, not statistically significant. For E and F, black asterisks show statistical significance between untreated and compound-treated *S. suis* P1/7 whereas red asterisks show statistical significance between untreated and compound-treated *S. suis* HE06.

Malaria Box (MMV667486 and MMV667487), and a compound from Pathogen Box (MMV675968), were tested for *Ss*DHFR inhibitory activity and *S. suis* growth inhibition. Pemetrexed, MMV667486, and MMV667487 did not show significant *Ss*DHFR or *S. suis* growth inhibitory activity (Figs. 2A, 2D and 2E), indicating that these structures do not inhibit *Ss*DHFR and consequently are unable to inhibit *S. suis* growth. Trimethoprim, pyrimethamine, and cycloguanil were moderately effective at inhibiting *Ss*DHFR and *S. suis* growth (defined as being effective at 10 µM but not at 1 µM) (Figs. 2B, 2D–2F). Methotrexate and MMV675968 inhibited *Ss*DHFR more effectively that trimethoprim, and completely inhibited *S. suis* growth, even at 1 µM (Figs. 2C–2F). Methotrexate and MMV675968, both of which are bicyclic 2,4-diaminopyrimidines with long and flexible

side chains, are highly effective inhibitors of both *Ss*DHFR and *S. suis* growth. Both *S. suis* P1/7 and *S. suis* HE06 strains shared a similar sensitivity pattern towards various DHFR inhibitors (Figs. 2E and 2F) consistent with the identical DHFR sequences between the two strains (Figs. S1A–S1B).

## DISCUSSION

Thirty compounds with high *S. suis* growth inhibitory activity were identified from the Pathogen Box library (Table S1). Among the thirty hits, seven are reference antibiotics known to be effective against Gram-positive bacteria, including levofloxacin, rifampicin, and linezolid (highlighted in blue in Table S1). The chemical structures of twenty three other hits are distinct from commercially available antibiotics (highlighted in green in Table S1). Approximately 60% of these hits (14 out of 23) also have activity against *Mycobacterium tuberculosis* (https://www.mmv.org/mmv-open/pathogen-box/about-pathogen-box), suggesting that they have broad activity against Gram-positive organisms and are attractive as starting points for developing antibiotics.

Among the 23 hits with novel structures, compounds MMV688508 and MMV687813 are oxazolidinone analogs. These compounds share the same core structure with radezolid, sutezolid, and linezolid, which are known to have broad spectrum activity against Gram-positive bacteria (*Bozdogan & Appelbaum, 2004*). Oxazolidinones inhibit bacterial protein synthesis by binding to the 50S ribosomal subunit and preventing the formation of a functional ribosome (*Bozdogan & Appelbaum, 2004*). It is therefore likely that MMV688508 and MMV687813 target *S. suis* protein synthesis in a similar fashion. Besides inhibiting the growth of *S. suis*, MMV688508 also inhibits the growth of other Gram-positive pathogens such as *Mycobacterium tuberculosis*, *Mycobacterium abscessus*, and *Staphylococcus aureus* (*Bhandari et al., 2018*; *Jeong et al., 2018*), showing broad-spectrum activity of MMV688508 against various Gram-positive pathogens. Interestingly, MMV687813 is not reported to have anti-*Staphylococcus* activity or anti-*M. abscessus* activity (*Bhandari et al., 2018*; *Jeong et al., 2018*) despite having activity against *M. tuberculosis* and *S. suis* (https://www.mmv.org/mmv-open/pathogen-box/about-pathogen-box, Table S1), pointing to structural differences among ribosomal targets and/or differences in compound permeability in various Gram-positive species.

DHFRs are present in various organisms including protozoan parasites, human, and bacteria. Even though DHFRs of different organisms perform the same function, i.e., the conversion of dihydrofolate (DHF) to tetrahydrofolate (THF) using NADPH as a cofactor, structures of DHFRs from different organisms are different and can be clustered into at least nine distinct clades, resulting in different druggable space and different structures of effective inhibitors (*Bhosle & Chandra, 2016*). DHFRs from various bacteria species do not necessarily cluster to the same clade. Specifically, *Ss*DHFR does not belong in the same clade as DHFRs of Gram-negative bacteria previously shown to be inhibited by MMV675968 (*Nelson & Rosowsky, 2001*; *Songsungthong et al., 2019*). Consequently, it cannot be assumed a priori that MMV675968 will effectively inhibit DHFR from *S. suis*. Experiment testing inhibition of *Ss*DHFR by MMV675968 was therefore needed.

This study is the first to show that MMV675968 has growth inhibitory activity against a Gram-positive bacterium, namely *S. suis*, by inhibiting *Ss*DHFR (Table S1, Figs. 1B and 1D), raising a possibility that MMV675968 may serve as a broad-spectrum antibiotic candidate. Growth inhibition by MMV675968 is reversed by thymidine supplement (Figs. 1C and 1E), confirming that the principal target of MMV675968 is a thymidine-related pathway.

Eight DHFR inhibitors of various chemical structures were tested for their efficacy in inhibiting *Ss*DHFR and *S. suis* growth, revealing structures associated with ineffective, moderately effective, and highly effective inhibitors (Fig. 2). Pemetrexed (a human DHFR inhibitor), MMV667486, and MMV667487 (*Plasmodium* DHFR inhibitors (*Aroonsri et al., 2016*)) were ineffective as inhibitors against *Ss*DHFR and *S. suis* (Figs. 2A, 2D and 2E). Trimethoprim, pyrimethamine, and cycloguanil, which are known inhibitors of bacterial and parasite DHFRs (*Gleckman, Blagg & Joubert, 1981*; *Rollo, 1955*; *Tonelli et al., 2017*), were moderately effective at inhibiting *Ss*DHFR and *S. suis* growth (Figs. 2B, 2D–2F). These moderately-effective inhibitors contain either diaminopyrimidine (trimethoprim and pyrimethamine) or diaminodimethyltriazine (cycloguanil) cores with a rigid or flexible side chain. Interestingly, even though MMV667486, MMV667487, and cycloguanil share the same diaminodimethyltriazine core and a phenyl ring at the same position, MMV667486 and MMV667487 were ineffective at inhibiting *Ss*DHFR and *S. suis* growth whereas cycloguanil was effective (Figs. 2A, 2B, 2D and 2E). The data suggest that the *Ss*DHFR active site can accommodate the shorter side chain of cycloguanil, whereas the longer/bulkier side chains of MMV667486 and MMV667487 may interfere with compound binding to the *Ss*DHFR active site, rendering the compounds ineffective as inhibitors. A correlation between *Ss*DHFR inhibitory activity and *S. suis* growth inhibition of various compounds was observed (Figs. 2D–2F), implying that the two *S. suis* strains are permeable to all DHFR inhibitors tested allowing access to the *Ss*DHFR target, leading to growth inhibition. It is possible that other *S. suis* strains may have different compound permeability patterns.

Among the inhibitors tested, MMV675968 and methotrexate were highly effective *Ss*DHFR and *S. suis* inhibitors (Figs. 2C–2F), both of which contain a bicyclic 2,4-diaminopyrimidine core with a long and flexible side chain. Such structures can serve as starting points for further rational drug design against *Ss*DHFR and *S. suis*. *Streptococcus pneumoniae* DHFR is approximately 53% identical to *Ss*DHFR and share high sequence homology at the folate and NADPH binding sites (Fig. S1C), raising a possibility that methotrexate and MMV675968 may also be effective against *S. pneumoniae* or other *Streptococci*. Other bicyclic 2,4-diaminopyrimidines were able to inhibit *Streptococcus mutans* growth (*Zhang et al., 2015*), confirming this hypothesis. Since methotrexate is a known inhibitor of human DHFR, whereas MMV675968 shows 59-83 fold selectivity for bacterial DHFRs over human DHFR (*Nelson & Rosowsky, 2001*; *Songsungthong et al., 2019*), and is not cytotoxic against human HL-60 and HepG2 cell lines (https://www.mmv.org/mmv-open/pathogen-box/about-pathogen-box), MMV675968 might be a better starting compound for antibiotic discovery.

## CONCLUSIONS

A screen of the Pathogen Box compound library leads to identification of 23 new bioactive compounds against *S. suis* (Table S1). MMV688508 and MMV687813, oxazolidinone analogs, are strong inhibitors of *S. suis* growth (Table S1). Since MMV688508 is shown to inhibit the growth of multiple Gram-positive species (*Bhandari et al., 2018*; *Jeong et al., 2018*), MMV688508 may be able to serve as an antibiotic candidate against Gram-positive pathogens. MMV675968, a bicyclic 2,-4 diaminopyrimidine with a long and flexible side chain, was identified as a potent inhibitor of *Ss*DHFR and of *S. suis* growth. MMV675968 may be able to serve as a broad-spectrum antibiotic candidate since it is shown to inhibit the growth of both Gram-positive and Gram-negative bacteria (*Nelson & Rosowsky, 2001*; *Songsungthong et al., 2019*).

## ACKNOWLEDGEMENTS

Pathogen Box and specific compounds in Pathogen Box and Malaria Box were obtained from Medicines for Malaria Venture. Plasmid pL0035 (MRA-850), contributed by Andrew P. Waters, was obtained through the NIH Biodefense and Emerging Infections Research Resources Repository Resources, NIAID, NIH. We thank Wildan Firdaus and Buppa Arechanajan for technical assistance, Penchit Chitnumsub for helpful comments, and Philip J. Shaw for suggestions on the manuscript.

### Funding

This research received no specific grant from any funding agency in the public, commercial, or not-for-profit sectors. Ubolsree Leartsakulpanich was supported by a research grant (P16-52034) from National Center for Genetic Engineering and Biotechnology (BIOTEC), National Science and Technology Development Agency (NSTDA). Suganya Yongkiettrakul was supported by a RI research grant (P16-51873) from Food Biotechnology Research Unit, BIOTEC. Potjanee Srimanote was supported by Thammasat University Fiscal year budget and Thailand Research Fund (TRF). The funders had no role in study design, data collection and analysis, decision to publish, or preparation of the manuscript.

### Grant Disclosures

The following grant information was disclosed by the authors:
National Center for Genetic Engineering and Biotechnology (BIOTEC).
National Science and Technology Development Agency (NSTDA).
Thammasat University Fiscal year budget.
Thailand Research Fund (TRF).

### Competing Interests

The authors declare there are no competing interests.
## Author Contributions

- Warangkhana Songsungthong conceived and designed the experiments, performed the experiments, analyzed the data, prepared figures and/or tables, authored or reviewed drafts of the paper, and approved the final draft.
- Sunisa Prasopporn, Louise Bohan performed the experiments, analyzed the data, prepared figures and/or tables, authored or reviewed drafts of the paper, and approved the final draft.
- Potjanee Srimanote, Ubolsree Leartsakulpanich and Suganya Yongkiettrakul conceived and designed the experiments, authored or reviewed drafts of the paper, and approved the final draft.

## Ethics

The following information was supplied relating to ethical approvals (i.e., approving body and any reference numbers):

BIOTEC Institutional Review Board on biosafety and biosecurity granted approval to carry out the study in its facilities (approval number BT-IBC-59-028).

## DNA Deposition

The following information was supplied regarding the deposition of DNA sequences:

Nucleotide sequences of the *dhfr* gene from *S. suis* P1/7 and *S. suis* HE06 are available at NCBI Genbank: MH388486 and MH388487, respectively

## Data Availability

The raw data are available in the Supplemental Files.

## Supplemental Information

Supplemental information for this article can be found online at http://dx.doi.org/10.7717/peerj.10743#supplemental-information.

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
