# Peer review of "A novel bicyclic 2,4-diaminopyrimidine inhibitor of Streptococcus suis dihydrofolate reductase"

_PeerJ, doi:10.7717/peerj.10743_

## Round 0.1 · original submission · Major Revisions

Reviewers #1 and #3 have important suggestions for additional experiments that would appreciably strengthen your manuscript.

Reviewer 1 ·

Basic reporting

In this study, the authors screened the open-source Pathogen Box compound library for new drug candidate against Streptococcus suis. They found a potential compound MMV675968 and tried to improve that it targets SsDHFR. The paper is clearly written and professional English language is used throughout. The structure conforms to PeerJ standards, and raw data are supplied. However, the background should be improved to show the study context better, and the Figures should also be improved.
For background:
1. In Abstract, the authors mentioned “S. suis is an emerging zoonotic pathogen” (line 21), which I agree, but in Introduction (line 46-49), it hard to tell the description is for human infection. Please clarify.
2. In Abstract, the authors said “it has become increasingly resistant to multiple classes of antibiotics” (line 22), but this statement lacks supporting materials throughout the article. Detailed evidences about “resistant to multiple classes of antibiotics” should be added somewhere around line 51 in Introduction.
For Figures:
1. The overall resolution should be improved, now both the Figure 1 and Figure 2 seem a little blurry.
2. Figure 1D, I think the title of the Y axis is not “percent growth”.
3. When you denotes the statistical significance with “*”, it is better to indicate which two sets of data you are compared.

Experimental design

The study is original primary research within Scope of the journal, most of the methods are described with sufficient detail & information to replicate, partial of the research question is well defined, relevant and meaningful, and some of the investigation are performed to a high technical & ethical standard. However, major improvements are needed.
1. The authors’ most important issue. You can not say the MMV675968 TARGETs SsDHFR without ruling out the possibility that the inhibition of S. suis growth is resulted from the inhibition of other important target by MMV675968. Thus, as you do for E. coli surrogate, at least you should make sure that the S. suis growth inhibition by MMV675968 can also be reversed in the presence of thymidine supplement. Only after confirmation of this, you can say MMV675968 targets SsDHFR, and then you can discuss about the permeability of DHFR inhibitors (in Discussion, line 229-234).
2. Minor questions: 1) Line 94, even you cited the E. coli surrogate you used in your study, it is better to shortly describe why both DHFR and TS enzymes were deleted; 2) Like you define the percent growth in line 80, you should also define the percent activity of DHFR in line 116; 3) Line 127, there are inhibitory activities of at least 3 compounds (No. 123, 269 and 383 in Table S1) are not 90% or greater inhibition of growth; 4) Line 148, from Figure 1B, the inhibition is not “completely”; 5) Line 163, where are the eight DHFR inhibitors from? If they are from Pathogen Box, please indicate their numbers; 6) Line 177 and line 179, please use different color in Table S1 to separately indicate the “eight” and “26” hits.

Validity of the findings

1. The novelty of the findings is not very convincing. The authors said MMV675968 is a novel inhibitor of S. suis dihydrofolate reductase. As we can see from the discussion, the authors has already found MMV675968 is a inhibitor for A. baumannii DHFR, is there significant difference between the DHFRs from S. suis and A. baumannii, or between DHFRs from gram positive and negative bacteria? If there is, please discuss. If there isn’t, I don’t think it is a NOVEL inhibitor for DHFR just because the bacterium is different.
2. In line 237-239, “can serve as broad-spectrum antibiotic candidate” “can serve as an antibiotic candidate against Gram-positive pathogens” are speculations but not conclusions, please change the tone correspondingly, at least do not be so affirmative.

Reviewer 2 ·

Basic reporting

1) In the introduction section the Authors have also to describe the role of DHFR as main target of different classes of compounds (i.e. antimalarial, antibacterial and antineoplastic agents [Targeting dihydrofolate reductase: Design, synthesis and biological evaluation of novel 6-substituted pyrrolo[2,3-d]pyrimidines as nonclassical antifolates and as potential antitumor agents. Eur J Med Chem 2019, 178, 329-340]).
2) They have also to cite the more recent role of host DHFR for anti-influenza drug development [Tonelli M., Naesens L., Gazzarrini S., Santucci M., Cichero E., Tasso B., Moroni A., Costi MP, Loddo R. Host dihydrofolate reductase (DHFR)-directed cycloguanil analogues endowed with activity against influenza virus and respiratory syncytial virus. Eur. J. Med. Chem. 2017, 135, 467-478].
3) In the title and also in the paper main text modify “2, 4 diaminopyrimidine” with “2,4- diaminopyrimidine”.
4) Besides citation 30 and 31 (row 210) add the previously indicated citation related to cycloguanil as antiviral agent [Tonelli M., Naesens L., Gazzarrini S., Santucci M., Cichero E., Tasso B., Moroni A., Costi MP, Loddo R. Host dihydrofolate reductase (DHFR)-directed cycloguanil analogues endowed with activity against influenza virus and respiratory syncytial virus. Eur. J. Med. Chem. 2017, 135, 467-478]
5) In the conclusion, row 236: modify “MMV675968, a bicyclic 2, 4 diaminopyrimidines” in ““MMV675968, a bicyclic 2,4-diaminopyrimidine”
6) Conclusions are too short, they should be more detailed.

Experimental design

'no comment'

Validity of the findings

'no comment'

Additional comments

The manuscript describes the discovery of a novel 2,4-diaminopyrimidine inhibitor, namely MMV675968, from a screening of the open-source Pathogen Box compound library against Streptococcus suis.
This compound have been evaluated in enzyme inhibition assays on the S. Suis DHFR and also in cell-based assays to assess growth inhibition of two S. suis isolates. Reversal effect on antibacterial activity has been demonstrated in S. suis-infected E. coli PA414 cells exposed to compound MMV675968 in combination with supplement of thymidine.
MMV675968 proved to be effective in both two in vitro assays, comparing favorably with methotrexate for inhibitory potency. Additionally, MMV675968 demonstrates a better selectivity profile for SsDHR than human DHFR, thus showing an improved safety profile with respect to methotrexate.
On the whole, the biological data look solid and significant.
The manucript is interesting and references are adequate and updated.

Reviewer 3 ·

Basic reporting

The manuscript from Songsungthong shows the inhibitor of the dihydrofolate reductase from Streptococcus suis by a 2,4-diaminopyrimidine from the pathogen box of MMV and attend the criteria of PeerJ judgment.

Experimental design

Although the results are interesting, in my point of view, other experiments could bring a more robust work to be published and cited. I am quite concerned about the methods used for in vitro activity of SsDHFR. Although the group has published other manuscripts using this protocol, I wonder why they did not perform the protein purification of the enzyme since they Ahave it already expressing in E. coli. I think more details bout the enzymatic assay should be add in methods since it is also not clear and I am assuming they are working with the absorbance of NADPH consume. If this is the case, even with the low chance, it could have the interference of other enzymes from E. coli to have some background effect as NADPH/NADP+ is a universal coenzyme and used for a number of different enzymatic systems. So in the way it the result is present could have a chance of the results to be misinterpreted. So, I strongly suggest that the authors use a pure enzyme instead of a cellular lysate.
In addition, although the authors suggest that MMV675968 is a “strong” inhibitor, I believe it could be considered a moderate inhibitor of the enzyme since they reported an inhibition at 10 µM. This although seems to be higher than TMP and other non-classic antifolates, in my opinion, is not a strong inhibitor.
Also, although the MMV has reported that these compounds are non-toxic, it should be interesting to have a toxicity experiment of MMV675968 against human cell lines or alternatively against the human enzyme to indicate the selectivity of the compound.
Finally, I also suggest using modeling tools to try to predict how this compound could bind in the active site of the protein. The DHFR from S. pneumoniae has about 54% of identity to S. suis and could be used even as a model to perform some molecular modeling of the enzyme and as well as docking simulation. A comparison between the active site could bring a plus to the manuscript.

Validity of the findings

Since the compound MMV675968 has a similar or even higher inhibition than other nonclassic antifolates, the findings are valid and interesting.

Additional comments

Interesting work, but quite preliminary and more experiments should be necessary to bring an advance in the field.

---

## Round 0.2 · Minor Revisions

Please address the remaining issues from Reviewer 1.

Reviewer 1 ·

Basic reporting

In this study, the authors found a potential compound MMV675968 and proved that it targets SsDHFR. Now the manucript has been greatly improved by modifing the background and Figures, and adding more necessary data. The paper is clearly written and professional English language is used throughout.

Experimental design

The research is within Aims and Scope of Peer J, and raw data are supplied. Research question is well defined, relevant and meaningful.

Validity of the findings

All underlying data have been provided. They are statiscally sound and controlled.

Additional comments

My comments are minor.
Line 144, for the definition of percent SsDHFR activity, it is better to use “A340 reduction” than “A340 change” to clarify better that you mean A340 reduction. Even it is reduction, the numbers still should be positive, which I noticed that you calculated as negetive numbers in Raw Data tables (1D and 2B), you just need to calculate reversely. Furthermore, I don’t think the calculted “A340 reduction/min” is raw data. I bring this issue up is because that the A340 reduction/min values (such as -0.08, -0.01, -0.0014) you calculated are too small, and they are even within the measurement error of a spectrophotometer. To make it more clear, you should mention the raw OD numbers in your raw data table and also mention the reaction time in your method.
Line 157, last time when I mentioned “there are inhibitory activities of at least 3 compounds (No. 123, 269 and 383 in Table S1) are not 90% or greater inhibition of growth”, I mean that even the value of percent growth is less than 10, however, it could be higher than 10 when you consider the SD, then the inhibition is less than 90% (not “90% or greater”, at least for one SS strain). Thus, its better to exclude these componds from the 34 “compounds with high inhibitory activity (with average percent inhibition of 90-100%)”.
Line 249, “DHFRs from various bacteria species do not necessarily cluster to the same clade. Consequently, even though MMV675968 is a known inhibitor of DHFRs of Gram-negative bacteria (Nelson & Rosowsky, 2001; Songsungthong et al., 2019), it cannot be assumed a priori that MMV675968 will effectively inhibit DHFR from S. suis.” Are the SsDHFR and the DHFR studied from the same clade? Which I assume not according to your description. If it is true, just clarify this and mention it is still necessary to test whether the MMV675968 is a effective inhibitor for SsDHFR.

Reviewer 3 ·

Basic reporting

I am happy with the corrections and the manuscript should be published.

Experimental design

fine

Validity of the findings

fine

Additional comments

fine

---

## Round 0.3 · accepted · Accept

Thank you for your thorough explanations of the few remaining issues. I am glad to accept your paper for publication.